**Subject Area:**
biochemistry

cellular parabiosis, cellular dysfunction, intercellular molecular traffic

**Author for correspondence:**
Miroslav Radman
e-mail: miroslav.radman@medils.hr

# Cellular parabiosis and the latency of age-related diseases

Miroslav Radman[1,2,3]

[1]Mediterranean Institute for Life Sciences (MedILS), 21000 Split, Croatia
[2]Naos Institute for Life Sciences, 13290 Aix-en-Provence, France
[3]Inserm u-1001, University R. Descartes Medical School, Cochin Site, 75014 Paris, France

Cellular parabiosis is tissue-based phenotypic suppression of cellular dysfunction by intercellular molecular traffic keeping initiated age-related diseases and conditions in long latency. Interruption of cellular parabiosis (e.g. by chronic inflammation) promotes the onset of initiated pathologies.

The stability of initiated latent cancers and other age-related diseases (ARD) hints to phenotypically silent genome alterations. I propose that latency in the onset of ageing and ARD is largely due to phenotypic suppression of cellular dysfunctions via molecular traffic among neighbouring cells. Intercellular trafficking ranges from the transfer of ions and metabolites (via gap junctions) to entire organelles (via tunnelling nanotubes). Any mechanism of cell-to-cell communication resulting in functional cross-complementation among the cells is called *cellular parabiosis*. Such 'cellular solidarity' creates tissue homeostasis by buffering defects and averaging cellular functions within the tissues. Chronic inflammation is known to (i) interrupt cellular parabiosis by the activity of extracellular proteases, (ii) activate dormant pathologies and (iii) shorten disease latency, as in tumour promotion and inflammaging. Variation in cellular parabiosis and protein oxidation can account for interspecies correlations between body mass, ARD latency and longevity. Now, prevention of ARD onset by phenotypic suppression, and healing by phenotypic reversion, become conceivable.

## 1. Introduction

'All things are difficult before they are easy' (Thomas Fuller, 1608–1661) and all things appear complicated before they become simple. This and the accompanying paper [1] propose simple concepts that can account for the aetiology of ageing and age-related diseases (ARD). Apparent complexity of ageing and diseases (actually, of their consequences) reflects the complexity of healthy organisms but reveals little to nothing about the root cause(s) of age-related morbidity and mortality. Identifying the causes and early cellular stages of ARD will be instrumental in eventual mitigation of degenerative diseases, including cancer. The answer to 'why' leads more productively to 'how' than vice versa. Spectacular low-tech success of vaccination and antibiotics in mitigation of infectious diseases (which are very complex at advanced stages) teaches that acting timely upon the cause can prevent or heal any disease. Hence, this paper and its companion deal only with the causation and early, pre-symptomatic stages of ARD and ageing.

The incidence of death and diverse ARD increases with about the fifth power of time, suggesting a common biological clock (with species-specific speed) and perhaps a shared common cause. A common cause would mean that it might be simpler to mitigate all ARD collectively than any particular one—reminiscent of the long-lasting simultaneous resistance of super-centenarians to all ARD.

Ageing is a process of biological degeneracy accelerating with age and resulting in ARD predisposed by inborn 'weak links'. To mitigate ARD, we

need to know the root cause of biological degeneracy and identify inborn weak links [1]. Ageing and disease phenotypes emerge after long periods of latency proportional to species' lifespan and body mass [2] proposed here to be contributed by *cellular parabiosis* (i.e. suppression of recessive and many dominant cellular phenotypes via molecular traffic among neighbouring cells; figures 2–5). Borrowing the terminology from experimental carcinogenesis, ARD appear to be stably *initiated*, probably by alterations in DNA sequence or modification, mainly as the consequence of a malfunction of damaged dysfunctional proteins dedicated to the integrity of somatic genomes [1,3] (figure 1). While cellular parabiosis can keep initiated diseases in a state of dormancy (figure 2 and figure 4), *promotion* triggers phenotypic expression (onset) of disease via interruption of cellular parabiosis (e.g. by chronic inflammation or by age-related extinction of phenotype-suppressing cells; figures 4 and 5).

The basic chemistry and genetics of the initiation of latent ARD and ageing are addressed in the companion paper [1]. Here, I explore the mechanism of promotion (i.e. phenotypic expression, or onset, of age-related pathologies). Plausible common cause [1] (figure 1) and a common mechanism of emergence (figures 2 and 4) of ageing and ARD phenotypes are integrated into a simple phenomenological model (figure 5) that could inspire promising approaches to active targeted prevention and even healing of ARD by their phenotypic reversion to health.

The snowballing consequences of ageing and ARD are complex and banal—a mere display of an organism's complexity—unrelated to the cause and nearly irrelevant to mitigation of ARD and ageing. Hence, this article considers only the early molecular and cellular stages of ARD and ageing when their onset can be prevented, arrested or even reversed. I present a phenomenological analysis of age-related pathologies unencumbered by descriptive molecular studies of the 'downstream' consequences (symptoms) of advanced diseases and ageing. The ideas and proposed concepts were inspired by the literature on biology of diseases and ageing spanning over seven decades. Hence, I refer often to adequate reviews.

## 2. Initiation of age-related diseases

In the companion paper [1], we argue that oxidative damage to imperfectly folded oxidation-sensitive proteins is a common root cause of ageing and ARD, and elaborate why protein damage, rather than DNA damage, is the leading cause of age-related genome alterations. Here, I recall the key concepts relevant to the initiation of latent ARD in order to propose a common mechanism of their phenotypic expression (i.e. onset of disease). The phenotype of degenerative diseases is linked to cell malfunction and death, whereas that of cancer involves unrestrained cell growth.

A common quantitative relationship between oxidative protein carbonylation induced by UV and ionizing radiation and killing of bacteria and metazoan germ line cells—regardless of huge differences in cellular radiation resistance—is shown in fig. 3 of [5]. Such common function for prokaryotic and reproductive eukaryotic cells is rare in biology and it leaves no space for any significant mechanism of cell death by radiation that does not involve protein

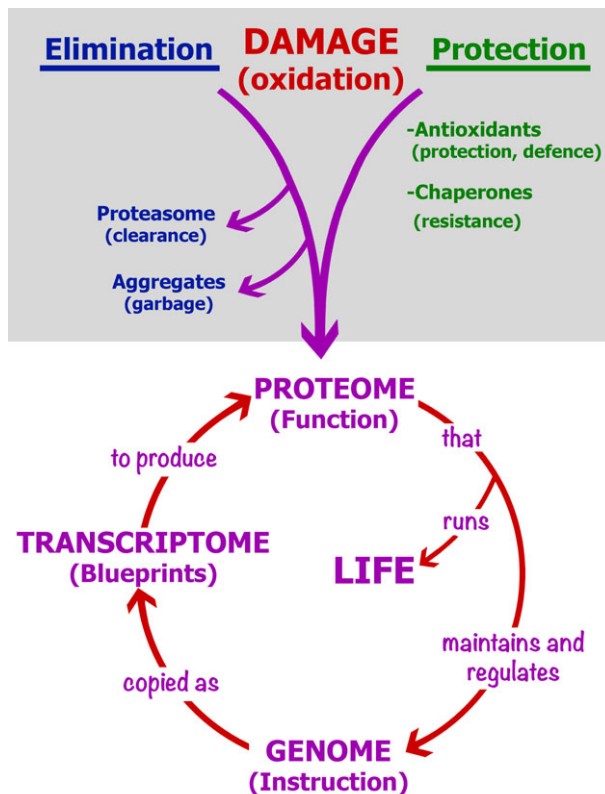

**Figure 1.** Oxidative protein damage and its impact. The circle in the lower part shows the obvious: protein damage can generate a wide variety of cell phenotypes, including genome alterations, eventually triggering a vicious circle of protein and DNA errors and consequent malfunctions (as in L. Orgel's error catastrophe theory [4]). The upper (shaded) area shows the reduction in potential proteome oxidative damage by passive protection (by antioxidants), active defence against ROS (by detoxifying enzymes) and increased intrinsic protein resistance to oxidation via native folding, as well as the removal of damaged proteins (in blue). 'Protection' (in green) includes evolved, structure-based protein resistance to oxidation (hence, the effect of chaperones) reduced or lost by silent protein polymorphisms (see text).

oxidation. A similar common function emerges when plotting protein carbonylation versus biological time (fraction of lifespan) of species as different as nematode, insect, rodent and human, including progerias [6] (reviewed in [1,7]). Thus, radiation and ageing kill cells in correlation with protein carbonylation, shown to be causal [1].

Irreparable protein damage such as oxidative carbonylation is far more mutagenic than reparable DNA damage [1,3] and can cause directly any kind of phenotypes, including cell death [1,5]. In *E. coli,* spontaneous and UV-induced mutation rates increase with about the seventh power of protein carbonylation, while mutations emerge linearly with respect to inflicted DNA damage [5,8]. Moreover, reducing exclusively the level of proteome carbonylation, at constant reactive oxygen species (ROS), reduces mutation rates about 10-fold below the wild-type level identifying oxidative protein damage as the principal determinant of spontaneous mutation rates [3] (figure 1).

Errors in the somatic maintenance of DNA methylation patterns, in particular, gene silencing involving hypermethylation of some specific CpG islands, are (i) age-related, (ii) more frequent than mutations, (iii) diagnostic of human biological age and (iv) predictive of remaining life [9,10]. Thus, DNA methylation appears as the overarching

royalsocietypublishing.org/journal/rsob    *Open Biol.* **9**: 180250

royalsocietypublishing.org/journal/rsob    Open Biol. 9: 180250

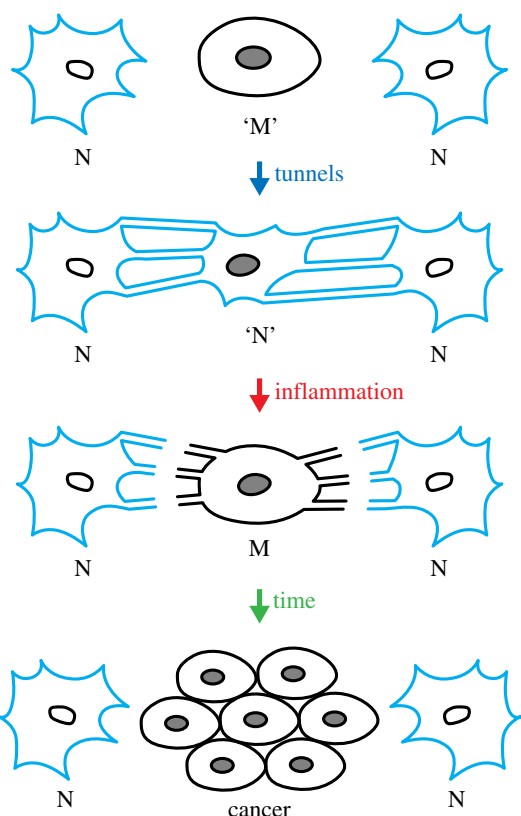

**Figure 2.** An illustration of phenotypic suppression of malignancy by cellular parabiosis, and phenotypic expression upon its interruption. Cellular parabiosis is the traffic of metabolites and functional and informational molecules between neighbouring cells preventing the expression of recessive phenotypes (see text). N are normal (blue) cells, 'M' is a premalignant cell (with dark nucleus) with suppressed phenotype ('N'), while M is a phenotypically malignant (with dark nucleus and black membrane) cell isolated from its normal neighbours by inflammation-mediated disruption of cell–cell connections. 'Tunnels' are tunnelling nanotubes or any other means of molecular traffic. A malignant cell, unsuppressed during chronic inflammation, can start dividing and become surrounded by isoproteomic sister cells precluding phenotypic suppression, allowing unrestrained malignant growth from the inside of the cellular monoclone that becomes a primary tumour ('cancer'). The same scheme applies to all other ARD where, instead of 'cancer', the phenotype is cell dysfunction or death.

candidate for genome-based initiation of cellular malfunction in human ageing.

While gene silencing by DNA methylation and gene inactivation by mutation display similar phenotypes, the incidence of somatic mutations [11] appears insufficient to account for an organism's ageing via loss of function or gain of toxic function in diploid somatic cells (except for cancer where single mutant cells can lead to lethal tumours). Akin to the aetiology of mutations, the leading cause of alterations in DNA and histone modification patterns must be the reduced efficacy and/or fidelity of proteins involved in the maintenance of DNA and histone modifications. Such malfunction is typical of the effects of protein damage [12], either directly or by affecting gene expression. This paraphrases the error catastrophe concept [4] in that, like all vital functions, genome stability and expression is a phenotype that depends directly on proteome quality (figure 1).

Proteomes of aerobic organisms have evolved a remarkable intrinsic resistance to oxidation, yet individual proteins display great variability in their intrinsic susceptibility to oxidative damage (fig. 6 in [1]). The fact that the intrinsic protein resistance to oxidation is fragile is highly relevant to ageing and diseases: random errors in sequence and folding, as well as phenotypically silent missense mutations (polymorphisms), greatly sensitize the affected proteins to oxidative damage [3,12,13] (see also fig. 7 in [1]). Misfolded protein structure is fixed (paralysed) by oxidative damage, therefore oxidation prevents phenotypic buffering of missense mutations by chaperones [1] (see also fig. 3 in [3]). Hence, some silent mutations appear as conditional mutations (i.e. they predispose to particular ARD by becoming progressively phenotypic with age/oxidation) [1].

Apparently, no major vital cellular function remains unaffected by ageing [14]. All molecular and cellular changes associated with ageing and ARD appear as snowballing consequences of increasing dysfunction of cells that accumulate oxidative protein damage such as irreparable carbonylation and reparable glycation (by DJ-1/PARK7 deglycase) [15]. Two major cellular systems appear prevalent in determining the level of protein damage: deactivation of mTOR pathway reduces the susceptibility of proteins to oxidation (via increased translational fidelity) and activation of FOXO pathway reduces the levels of ROS (via Nrf2-dependent ROS detoxification) [1]. Furthermore, both conditions stimulate the clearance of damaged aggregated proteins. While strategies for mitigation of ARD at the level of their cause (disease initiation) are explored in [1], the mechanisms of latency and onset of diseases are analysed below.

## 3. Promotion of diseases: latency and cellular parabiosis

The notion of parabiosis (living by proximity) derives from an experiment by Paul Bert in 1864 that revealed physiological effects of shared bloodstream between two surgically joined animals [16]. Over a century later, 'heterochronic parabiosis' between isogenic old and young mice showed rejuvenation—and even healing of diseases—lasting only during ongoing parabiosis [17]. Thus, parabiosis is phenotypic suppression, but not correction, of the initiated cellular defect. Here, I apply the term *cellular parabiosis* for the compensation of functional deficiencies of individual cells in tissues by intercellular communication via molecular traffic. The result of such phenotypic cross-complementation is the homogenization, or averaging, of cellular functions across the tissue, called tissue homeostasis. Let us consider the expected impact of cellular parabiosis on ageing and diseases.

A whale that can live 210 years has a million times more cells at risk of cancer than a mouse, yet lives cancer-free for a hundred times longer than a laboratory mouse (lifespan 2.5 years). This is referred to as Peto's paradox [18]. On a per cell and per unit time basis, there is an incomparably (at least $10^8$ times) lower incidence of cancer and other deadly diseases in whales than in mice. What is the difference between the biology of a whale, or human, cell *in vivo* and a similar cell in a mouse or shrew? Apparently, whale cells have evolved ways to reduce the initiation and/or prolong the latency of ageing and diseases. If we knew how, we could use that knowledge to mimic, healthwise, the biology of virtual huge humans to mitigate diseases and extend lifespan in a manner already developed during evolution. If

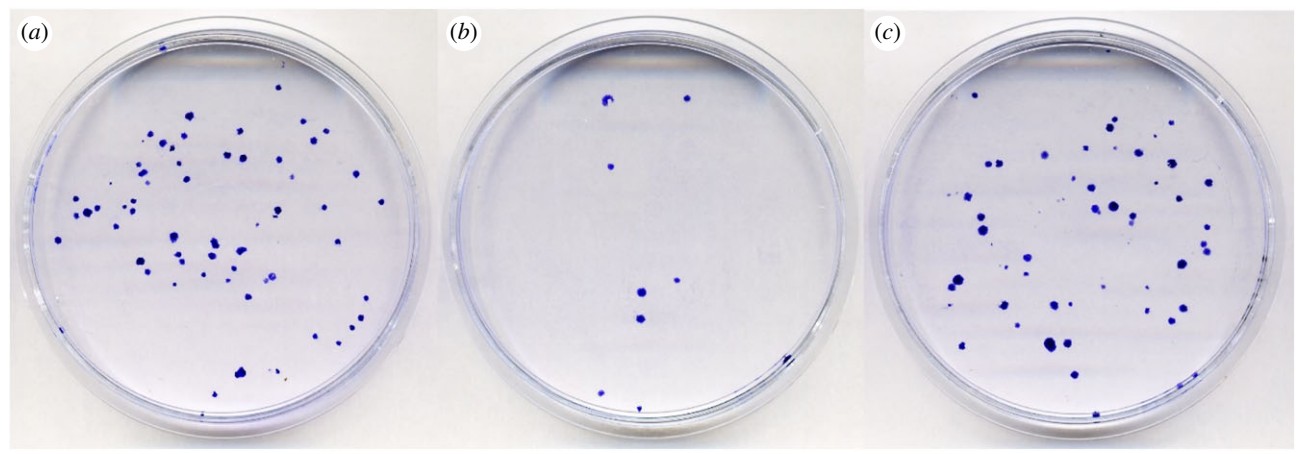

**Figure 3.** The Newbold–Amos experiment [26] reproduced to visualize the phenotypic suppression of recessive 6-thioguanine resistance (6-TG^R) by the proximity of wild-type cells, and its prevention by the inflammatory agent, the tumour promoter TPA/PMA. Since 6-TG resistance is the phenotype of a loss-of-function mutation (Hprt enzyme deficiency), this figure displays symbolically a 'disease in the Petri dish' (by the number of growing blue Hprt⁻6-TG^R mutant colonies) phenotypically suppressed by the excess of normal cells (compare (a,b)) and activated (promoted) by TPA/PMA (compare (b,c)). See the text and [26]. (a–c) plates were seeded each with 100 V79 Hprt⁻6-TG^R cells. (b,c) received in addition $7.5 \times 10^5$ V79 Hprt⁺ 6-TG^S cells. (c) contains also 1 ng ml⁻¹ TPA/PMA. 6-TG and where appropriate TPA/PMA were added together 4 h after seeding and the cells fixed after 72 h incubation, then stained by crystal violet. (a–c) plates contain 68, 9 and 62 colonies, respectively. (b) shows 87% efficient phenotypic suppression that is nearly fully (91%) prevented by TPA/PMA in (c). The nine 6-TG^R colonies in (b) grew presumably because of a delayed parabiotic contact with growing 6-TG^S Hprt⁺ cells. (Courtesy of Sara Trifunovic and Mila Ljujic.)

the principles of a medicine for small animals can be found in large animals, then we should try to understand them in order to slow down ageing and mitigate ARD. Below, the description of a concept for the latency of ARD and ageing is followed by chapters integrating the current knowledge within the framework of the concept.

## 3.1. Phenotypic expression and suppression of age-related pathologies

Among random phenotypic mutations, most are recessive (i.e. due to the reduction or loss of function that, when affecting regulation, can lead downstream to a gain of function or dysfunction). Some malignant and senescence phenotypes appear recessive (i.e. suppressed) in cell fusion experiments (reviewed in [19]). Chromosome shedding during the divisions of tetraploid cell fusion products causes such suppressed phenotypes to reoccur. Clearer evidence for a lasting phenotypic suppression is provided by direct cell–cell contact: malignant growth of individual tumour cells can be suppressed in two-dimensional (2D) cell cultures simply, and strictly, by direct contact with a monolayer of normal cells (e.g. fibroblasts) [19–21]. Some cell types that do not express proteomes adequate for phenotypic suppression will not suppress malignancy [19].

Phenotypic expression of newly acquired recessive genome alterations emerges in solitary cells with the fading out of residual (non-renewing) protein activity either by protein dilution (if the cell divides) or by protein damage, aggregation or proteolysis (in quiescent and in dividing cells). However, for cells within the tissues, or confluent cell cultures, recessive and some dominant alterations can remain phenotypically silent for an indeterminate time [21]. The larger the animal, the longer the latency in the onset of ageing and disease. How can an animal's size affect phenotypic expression of disease-initiating genome alterations in single cells (i.e. which cellular functions are affected by the body mass)?

Based on published data, I propose the existence of a lasting phenotypic suppression, even phenotypic reversion, of cellular deficiencies by means of known (and perhaps also unknown) mechanisms for the acquisition of metabolites, as well as functional and informational macromolecules, and entire organelles, transferred and shared among neighbouring cells [22–25] (figures 2 and 4). Molecular trafficking between cells can be function-compensating by metabolic cooperation—known to prevent phenotypic expression of newly induced recessive mutations—by plating mutagenized cells at high density ([26] and references therein). Even in growing cell cultures, it can take up to 9 days to reveal all induced recessive mutations [27]. Furthermore, the already expressed mutations are subject to phenotypic suppression/reversion (e.g. reversible loss of recessive drug resistance) by plating mutant cells with an excess of wild-type cells at high density [26] (figure 3).

Many names, often redundant, were given to diverse extracellular structures shown, or presumed, to be the vectors of intercellular molecular traffic [22–25]. Most familiar are gap junctions, desmosomes and neuronal synapses that allow for the trafficking of metabolites and electrolytes. Since the majority of cellular catalytic activities are involved in the biosynthesis of metabolites and their intermediates (at least 120 000 kinds in humans), gap junctional traffic may phenotypically suppress the majority of random genome alterations. However, tunnelling nanotubes (TnT) [22,25], extracellular vesicles [23] and/or exosomes [24] were shown to transfer diverse RNAs, proteins (including prions) and even entire organelles (mitochondria, lysosomes, Golgi apparatus) and viruses—all functionally expressed in recipient cells [22,23]. One study showed unequivocally that dozens of protein species pass directly between the cells through TnTs [28]. The most efficient means for molecular exchange between cells packed in tissues may be still undiscovered (e.g. imaginable recurrent partial cell fusions). The presence of diffusible effectors of cellular parabiosis in conditioned media would not be surprising [29].

**Figure 4.** Expression of latent cellular defects by inflammation and senescence-associated secretory phenotype (SASP) via interrupted cellular parabiosis. The hexagons are individual cells; open circles are nuclei with fully functional genomes; black circles are nuclei with genomes bearing recessive genetic or epigenetic DNA alterations; shaded cells express their recessive phenotypes due to failing cellular parabiosis. Inhibiting inflammation and SASP turns the arrow in opposite direction by the re-establishment of parabiosis and phenotypic suppression (i.e. healing by phenotypic reversion).

For the sake of argument, it suffices to recall that cellular contents ranging from ions and metabolites to entire mitochondria are subject to intercellular trafficking generating phenotypic suppression of the recipient cell's deficiency (e.g. restoration of respiration proficiency by mitochondrial transfer into respiration-deficient cells [30]). TnTs display curious directionality: they grow from radiation or hydrogen peroxide-damaged cells towards intact cells [25], and mitochondrial transfer is apparently directed from respiration-proficient towards deficient cells [30]. If metabolic stress induces growth of TnTs (seeking help), and the F-actin polarity of TnT structures imposes directionality of molecular and organelle transfer opposite to TnT growth, then all TnTs may transfer unidirectionally. Direction of transfer between two cells will appear bidirectional when TnTs extend from both partner cells, and unidirectional when TnTs extend from only one partner.

A massive molecular trafficking among the cells *in vivo* is expected to functionally homogenize entire tissues and organs via dominant-positive effects on non-overlapping cellular deficiencies (figure 4). Such molecular traffic is expected to keep initiated cancer and other age-related pathologies (e.g. cardiovascular, neurodegenerative, immunological, type II diabetes, rheumatoid arthritis, etc.) in the state of latency by functioning as a 'cellular parabiosis' network. I propose this name to encompass any mechanistic variety of tissue-based suppression of deleterious phenotypes. Phenotypic suppression of somatically acquired deficiencies via cellular parabiosis offers a physiological and mechanistic meaning to the term tissue homeostasis. With cellular parabiosis, one can consider entire tissues and organs functioning (to a limited degree) as a single gigantic cell with billions of genomes, reminiscent of multinucleate muscle cells, which rarely, if ever, become malignant while their mononuclear progenitor cells do.

In a remarkable review, Harry Rubin [21] presented a wealth of observations about phenotypic suppression and expression *in vivo* versus in *ex vivo* primary cell cultures that can be readily interpreted by, and lend support to, the cellular parabiosis hypothesis. Rubin argued that a limited number of successive cell divisions (Hayflick limit) is nearly 100 times greater *in vivo* (human tissues) than *in vitro* (cell cultures), suggesting that telomere maintenance in tissues—a bottleneck to cell division—might take advantage of cellular parabiosis.

Undesirable effects of cellular parabiosis upon health, such as increased fitness of tumour cells and cell-to-cell transfer of infectious particles, such as viruses, bacteria and prions, have been put forward in relation to TnTs [22,25]. Indeed, aggressiveness and radiation resistance of human gliomas (astrocytomas) is causally related to the capacity of tumour cells to grow networks of intercellular TnT-like connections within the tumour [31,32]. This demonstrates the contribution of cellular parabiosis to the fitness of a developed tumour that was, presumably, initially kept in long latency by parabiosis with normal cells [32]. I posit that this latter positive, ageing and disease-preventing effect of cellular parabiosis in the early phases of ARD (at single to few cells level) provided an evolutionary advantage by extending the parental longevity required for reproduction of large organisms.

The proposed cellular parabiosis (figure 2) could explain why primary tumours generally do not metastasize within the same organ—for the same reason (phenotypic suppression of malignancy) responsible for long latency in the onset of the primary tumour. Presumably, a tumour cell translocated within the same organ would have to undergo the same kind of long phenotypic latency as the initial primary tumour cell before growing into a metastasis. This idea generates specific predictions about organ selectivity of cancer metastases. Optimal homing organs should be either unable to establish parabiosis with intruding tumour cell, or better, not express

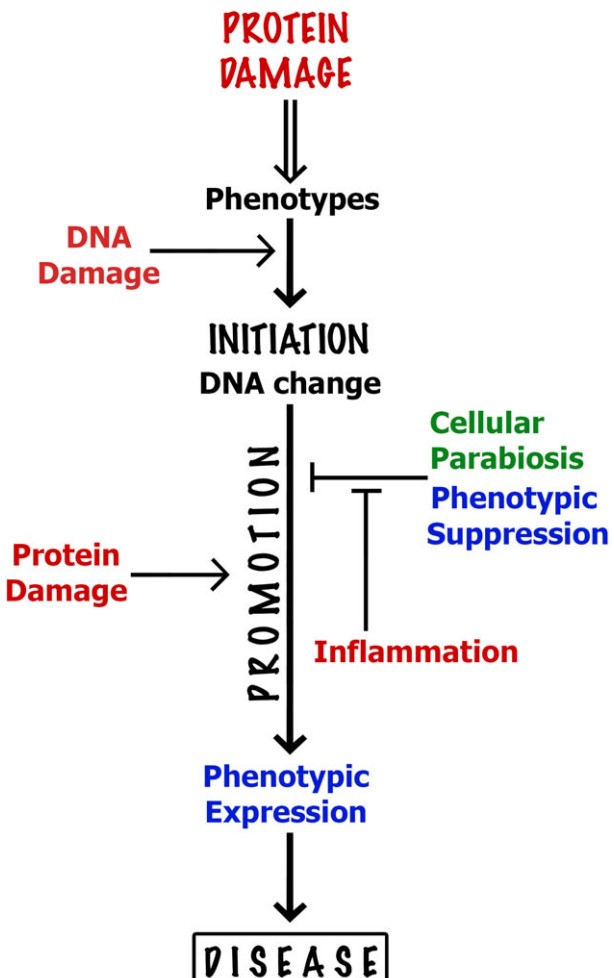

**Figure 5.** Key molecular and cellular processes affecting the onset of ageing and age-related diseases (e.g. carcinomas). 'Initiation' and 'promotion' are two key mechanistic phases in the onset of ARD. Oxidative proteome damage by ROS displays diverse reversible phenotypic effects, including generation of genetic and epigenetic alterations ('initiation' by DNA change) kept in phenotypic silence by cellular parabiosis (phenotypic suppression). 'Promotion' is typically a chronic inflammation that interrupts cellular parabiosis (see text and figure 5) leading to phenotypic expression, i.e. the onset of disease. DNA damage merely synergises generation of (epi)mutations since protein damage alone is more mutagenic than DNA damage [1]. Lasting promotion via chronic inflammation activates initiated morbid phenotypes ranging from malignant growth to cell death. Red coloured processes produce, or are affected by, ROS and stimulate initiation and/or promotion of pathological phenotypes, whereas green and blue belong to tissue homeostasis. High ATP levels and low ROS increase proteome quality and promote health and longevity (see text).

tumour suppressors of the primary tumour's tissue while providing for compensation of all other metabolic deficits. Consequently, the suppression of malignancy by cellular parabiosis should be most efficient in isophilic tissue (with the same expressed proteome), while metastases will grow best in heterophilic tissues (not synthesizing the molecules that suppress the malignant phenotype). This interpretation of the organ specificity of metastases is being tested.

## 3.2. Inflammation as promoter of tumours and other age-related pathologies: a parabiotic viewpoint

Perhaps the most informative phenomenological studies on cancer are the 70-year-old initiation–promotion experiments on mouse skin by Isaac Berenblum and colleagues [33].

They showed that chronic 'sterile' inflammation by repeated application of a non-carcinogenic inflammatory skin irritant (croton plant oil containing a diterpene ester, tetradecanoyl-12-phorbol-myristate-13-acetate, TPA or PMA, as active agent) dramatically shortens cancer latency and increases the number of tumours initiated by single application of a mutagenic carcinogen [33] (reviewed in [19,34]). Thus, initiation induces latent tumours that can be subsequently activated by promotion equally efficiently at any point in life. Promotion must follow initiation: it has no effect if applied before initiation in a young animal: initiation appears as 'silent' stable (genetic or epigenetic) genome alteration phenotypically expressed by promotion (i.e. inflammation) [19,34].

Subsequently, it was shown that promotion is a cellular phenomenon, since TPA promotes malignant cell transformation initiated by radiation [35] and interrupts communication among cells in culture [36–38] (figure 2). This indicates that certain effects of inflammation can be observed in a Petri dish in the absence of infectious agents and immune system. To test for the effect of inflammatory agents on phenotypic expression of a loss-of-function mutation, we engaged in an experiment called colloquially 'disease in a Petri dish' (where the loss of *hprt* gene function, bestowing a drug resistance, was considered a 'disease') just to find out that precisely that experiment was published in 1981 by Newbold & Amos [26] with a spectacular result (pictured in figure 3 from a repeated experiment).

Nanomolar concentration of TPA, devoid of detectable toxicity, prevents a very efficient phenotypic reversion of drug (8-azaguanine and 6-thioguanine) resistance in dense co-cultures with the drug-sensitive Hprt$^+$ cells. Figure 3 was a reconstruction experiment in the Newbold & Amos paper [26] where they showed the lack of phenotypic expression of new 8-azaguanine-resistant mutants when mutagen-treated V79 cells were seeded at high density. Under such conditions, the addition of nanomolar TPA to growth medium led to a rapid expression of newly induced drug resistance mutations [26]. Thus, TPA alone promotes the expression of recessive mutations [26], as well as of malignant phenotypes [35] in confluent cell cultures. Whereas TPA induces canonical transcriptional cascades of inflammation and excretion of inflammation-associated proteases [39], not all effects of TPA need to be necessary in the promotion of disease (below).

Inflammation *in vivo* involves induced generation of ROS by the macrophages (to kill bacteria) and excretion of matrix metalloproteinases (MMPs) and some other serine proteases (e.g. plasminogen activators and plasmin) [39]. Presumably, such proteases break down parabiotic structures in the extracellular matrix to clear the way for the infiltration of macrophages and neutrophils to the site of infection. If long-lasting, inflammation-mediated cessation of cellular parabiosis will cause collateral damage to the affected tissue by promoting phenotypic expression of initiated cellular pathologies.

Therefore, specific inhibitors of tissue-specific MMPs, called TIMPs, get involved and limit the time of inflammation. But during chronic inflammation, cells become durably isolated *in situ* and therefore start expressing their genome-based defects (e.g. in tumour suppressor genes, accounting for tumour promotion in figure 2). Separation of either normal or malignant cells within inflamed tissues was documented by identifying their high counts in patients' bloodstream [40,41]. Unlike acute pathogen-associated inflammation, so-called sterile inflammation triggered by the

royalsocietypublishing.org/journal/rsob  *Open Biol.* **9**: 180250

royalsocietypublishing.org/journal/rsob    Open Biol. 9: 180250

recognition of 'non-self' entities can become chronic (i.e. lasting as long as these 'non-self' entities). Increased sterile inflammation is among the hallmarks of ageing associated with rheumatoid conditions. Extracellular 'self' proteins, when misfolded and oxidatively damaged, as in ageing, might be recognized by the immune system as 'non-self' and trigger age-related chronic inflammation (ongoing experiments).

However, limited sterile inflammation can display significant positive effects [42]; for instance, by acting as an extracellular matrix clean-up system by the clearance of abnormal aggregated 'non-self' proteins (e.g. amyloids, fibrins, keratins, elastins, etc.) thereby preventing their extracellular deposition and immune stimulation. Pulses of short-term reversible inflammation can clear extracellular protein garbage via activated MMPs and plasmin proteases (controlled by TIMPs and tPA/serpins/PAIs, respectively) and therefore benefit tissue homeostasis by facilitating the re-establishment of unencumbered cellular parabiosis. Hence, there is no conflict between the opposing effects of inflammation—acute versus chronic.

Somewhat related to sterile inflammation is cellular senescence-associated secretory phenotype (SASP) displayed by non-dividing senescent cells bearing damaged or telomere-depleted chromosomes [43]. Senescent cells that escape apoptosis accumulate with age and exert diffusible deleterious effects upon the tissues, akin to inflammation. Pathogenic potency of SASP was demonstrated by inducing selective apoptosis of SASP-expressing cells in mice, which resulted in improved health and extension of lifespan [44]. The spread of tumour-promoting effect of SASP occurs particularly to old tissues, bearing more genomic alterations than young ones [45]—presumably by disrupting cellular parabiosis since SASP is fully inhibited by specific inhibitors of the metalloproteinases MMP-1 and MMP-2 [45].

Tumours that, like senescent cells, exert SASP-like effects including MMP excretion display poor clinical prognosis [46]—probably by precluding their cellular parabiosis with potential malignancy-suppressing cells. Consistently, MMP inhibitors and microbial small protease inhibitors (e.g. antipain and leupeptin), originally isolated as anti-inflammatory agents, show both anti-promoting and direct anti-carcinogenic activities on mouse skin (reviewed in [47]). They could inspire the design of drugs with the specificity of TIMPs to target only MMPs in particular inflamed tissues, while allowing for infection-fighting inflammation in other tissues. This concerns the risk of the use of all systemic anti-inflammatory drugs since clinical trials (CANTOS) with effective generalized anti-inflammatory treatment (Canakinumab), while reducing cancer and cardiac pathologies, caused serious morbidity and mortality by the increase in infectious diseases [48].

In conclusion, inflammation appears as 'phenotypic expressor' (i.e. promoter of the onset of tumours and other ARDs, presumably via prevention or interruption of cellular parabiosis; figures 2–5). This goes beyond hypothesizing, since the potent inflammatory and tumour-promoting agent TPA was shown to activate the expression of a recessive drug resistance phenotypically suppressed in dense two-dimensional co-cultures with isogenic drug-sensitive (wild-type) cells [26] (figure 3). Since there is no immunity in this simple experiment, cryptic inflammation in Petri dish (perhaps just MMPs) may mimic promotion of initiated diseases (e.g. carcinomas). However, much remains ill-understood since the expression of malignant phenotype in cell culture requires confluency [49].

## 3.3. ROS, tissue homeostasis and inflammation in ageing and disease

Burtner & Kennedy [50] identified ROS and tissue homeostasis as two sole elements that are common to ageing in invertebrates, mammals and human progerias—and they are two centrepieces in the ageing and disease scenario proposed here and in the accompanying paper [1]. Protein damage by ROS appears as an overarching causal factor in both initiation (DNA alteration) and expression (reduced cellular parabiosis) of diseases (figure 5) [1]). Anti-inflammatory agents are also antioxidants and all inflammatory conditions and agents act as tissue oxidants. Key rate-limiting steps, or processes, in the onset of ARD, in particular carcinomas, are summarized in figure 5.

## 3.4. Homeostasis in extracellular matrix: protein aggregates *versus* hyaluronan

Maintenance of optimal conditions for cellular parabiosis brings to focus the hygiene of the extracellular matrix as the medium of cell–cell interactions. The accumulation of extracellular amyloid, fibrin or keratin aggregates—one of the hallmarks of ageing tissues seen prominently in neurodegenerative diseases—can isolate cells from their neighbours either directly (as physical barriers) or by instigating an inflammatory immune reaction. Moreover, some extracellular and intracellular misfolded and oxidized proteins form small oligomeric pore-like hydrophobic structures that kill cells when inserting into the membranes by causing fatal leakage of ions and metabolites [1,51]. Such oligomeric aggregates are the likely agents of massive cell loss as the principal cause of degenerative diseases (e.g. the neuronal death in Alzheimer's and Parkinson's diseases).

On the other hand, abundant extracellular high molecular weight hyaluronan is required for notorious cancer resistance in the naked mole rat [52]. hyaluronon can act as a viscous matrix protecting 'lubricant' whereas its interactions with CD44 receptor and TSG6 lead to potent suppression of inflammation [53]. Being the milieu of cellular parabiosis and the space housing diverse cytotoxic proteins, the extracellular matrix becomes a potential pharmacological target for the extension of functional longevity of the tissues via cellular parabiosis, particularly accessible in dermatology and skin care.

## 3.5. Tissue architecture, inflammation and promotion of pathologies

Molecular trafficking among cells is proposed to: (i) keep cells in metabolic and developmental synchrony, (ii) induce trans-differentiation of intruding heterophilic cells and (iii) keep malignant and other cellular defects in the state of phenotypic dormancy [21]. The arrangement of cell types in epithelia can serve to prevent expansion of altered cellular monoclones (bottom drawing in figure 2). Otherwise, even normal cell divisions could produce altered monoclones—morphologically distinct cell patches often seen in ageing organs—precluding phenotypic suppression within the isoproteomic monoclone.

As pointed out by John Cairns [34], the basement stem cell monolayer in epithelia assures segregation of daughter cells committed to differentiation and disposal as one way of

avoiding the outgrowth of (pre)malignant cell clones. Disordered epithelial architecture accompanied by morphological cell alterations is produced by MMP activity during inflammation [54] and is called epithelial–mesenchymal transition (EMT). EMT is normally required for restructuring tissues during development and wound healing, but it is also mechanistically associated with the emergence of carcinomas and their metastases [54].

## 3.6. Tissue-based mitigation of ageing and diseases

Whatever the detailed mechanism, some type of tissue-based phenotypic suppression such as cellular parabiosis is a logical necessity to account for the phenomenology of ARD [18,19,21,50], notably Peto's paradox [18,55] and experimental mouse skin carcinogenesis by initiation and promotion [33,34]. First evidence for the suppression of already expressed malignant phenotypes (i.e. phenotypic reversion) mediated by direct cell–cell contact is Michael Stocker's demonstration of growth suppression of single tumour cells deposited on a monolayer of normal fibroblasts (while solitary cells of the same tumour can grow, and lumps of tumour cells grow also on fibroblast monolayers) [20].

Cellular age-related phenotypic heterogeneity (including morphological, malignant and lethal phenotypes) displayed by culturing primary hepatocytes from old (but not young) mouse liver, can be viewed as phenotypic expression—upon the exit from *in vivo* cellular parabiosis—of stable pathological alterations accumulated with age (reviewed in [21,33] and pictured in figure 4).

To relate body mass and disease latency to cellular parabiosis, it would help to know about the diffusion of phenotypic suppression across multiple layers of cells in tissues. Compared with tiny animals, large body mass and cell number should provide for longer-lasting cellular parabiosis potential. Yet why should human organs not behave as an ensemble of hundreds of mouse organs? The bottleneck to effective cellular parabiosis, ARD latency and lifespan must be the rate of age-related extinction of phenotype-suppressing cells, presumably related to the consequences of accumulated oxidative damage. Mechanistically, body mass-related reduction in ROS generation [2], and therefore in protein damage, should assure better maintenance of protein activities dedicated to protein biosynthesis, DNA repair, replication and modification [1] resulting in lower rates of ARD initiation and longer-lasting parabiotic potential (figures 4 and 5). A hint for the former can be seen in the reduction of *ex vivo* cellular mutation rates with an increase in body size (10-fold from mouse to whale—sufficient to account for the differences in cancer rates [18]). Since reduced protein oxidation will reduce both initiation and promotion of ARD, ROS and protein susceptibility to ROS appear as principal root causes of ageing and disease (figure 5).

## 3.7. Cellular parabiosis and the inevitability of somatic death

The number of fit suppressing cells in tissues must be decreasing with ageing and stress that damage proteins and, consequently, modify genomes [1,3] (figures 1, 4 and 5). Therefore, ageing should and does progressively increase cellular interdependence in tissues expected from cross-complementation between cells bearing non-overlapping epigenetic and genetic defects. Thus, when separated in primary cell cultures [21], or *in situ* during prolonged inflammation, cross-complementing cells will express their morbid phenotypes. This was documented as age-related phenotypic heterogeneity (e.g. cellular morphology, death and malignancy) revealed during the establishment of primary hepatocyte cultures derived from the liver of old versus young mice (reviewed in [21]).

Figure 4 illustrates the trend of extinction of phenotype-suppressing cells with age: surrounded by accumulating defective (but functionally compensated) cells, fittest cells cannot amplify. Thus, while extending lifespan, the avoidance of counter-selection of defective cells due to cellular parabiosis assures eventual extinction of the entire parabiotic network and the certainty of death.

Poor viability and quality of cells in primary cultures from old organs [21] suggest that promotion (phenotypic expression) is the bottleneck to the onset of degenerative diseases. In that case, to mitigate ARD, targeting promotion (i.e. the improvement of phenotypic suppression) becomes a priority.

In contrast to somatic tissues, the immortality of germline cells and of malignant cells surviving passages in cell culture (without phenotypic suppression) is probably assured by constant selective amplification (overgrowth) of fittest autonomously growing cells. This is akin to the periodic selection of fastest growing sub-clones in continuous bacterial cultures. Such selection for fitness occurs with circulating primary tumour cells that undergo massive apoptotic death (anoikis) while few metastasize [54]. Similarly, short latency in the onset of leukaemias (compared to solid tumours, e.g. after Hiroshima and Nagasaki atomic bomb explosions) could be due to the absence of tumour-suppressing cellular parabiosis for circulating cells.

## 3.8. Parabiosis and the reversibility of ageing and diseases

A basic prediction of the proposed protein-centred model for ageing and ARD is the reversibility of deleterious phenotypes upon proteome renewal (except for acquired stable genomic alterations). Animal heterochronic parabiosis (shared bloodstream between isogenic old and young mice) shows rapid reversion of tested ageing and ARD phenotypes [17]. But the return to initial biological age upon interruption of parabiosis suggests that animal blood-mediated parabiosis (just like cellular parabiosis) acts by compensation, not correction, of age-related defects. Apparently, the cause of ageing and ARD initiation remains 'memorized', presumably in the form of altered DNA methylation and/or mutations both unaffected by heterochronic parabiosis. Testing DNA methylation pattern and mTOR status during heterochronic parabiosis, particularly in stem cells, would be instructive.

## 3.9. The evolution of health and life spans: metabolism, proteostasis, parabiosis

If the generation of ROS, the cellular management of ROS-mediated proteome damage and the efficacy of cellular parabiosis together tune the somatic life clock, all of them should

royalsocietypublishing.org/journal/rsob Open Biol. 9: 180250

royalsocietypublishing.org/journal/rsob    Open Biol. 9: 180250

be affected by the rate of oxidative metabolism, i.e. generation of ATP and ROS. ATP is the bottleneck in proteome renewal and chaperone activity, whereas ROS are the killers of proteome function [1]. Hence, the metabolic rate affects the trade-off between efficacy (reproduction, speed, etc.) and robustness (longevity). Oxidative phosphorylation is slow but efficient in producing ATP as well as ROS, whereas aerobic and anaerobic glycolysis, adopted by cancer and stem (ES and iPS) cells, is inefficient (8 and 16 times less ATP per glucose, respectively) but rapid with low ROS generation [56]. Indeed, of all eukaryotic cells tested, ES cells display by far the lowest measured protein carbonylation levels (tab. 1 in [1]).

Accurate protein biosynthesis and chaperone-assisted folding are the cell's highest energy (ATP) consumers (over 75%). Therefore, selective pressure for maintaining high ATP production, along with increased defence against damage by ROS, will cease once reproduction is assured. The larger and more complex the animal, the longer the time required for its reproduction and the longer lasting must be the high-energy investment in parental proteome quality. Indeed, there is a remarkable interspecies correlation between the increase in longevity and body mass on one hand, and the decrease in the rates of translational errors [57] and oxidative metabolism [2] on the other hand. Both decreasing rates reduce the rate of proteome oxidation (reviewed in [1]).

Clearly, the increased complexity and body mass of newborn organisms put selective pressure upon the mechanisms assuring parental longevity required for the perpetuation of species. Experimentally imposed delay in reproduction readily results in the selection for increased longevity of Drosophila [58]. The extension of longevity involves a reduction in cellular oxidative metabolism and in translational errors usually via IIS [insulin/IGF (insulin-like growth factor)-like signalling] and TOR/FOXO pathways [58,59] and generally correlates with increased species' body mass [2]. Even the 'outliers' in the body mass–longevity relationship support specifically this line of thinking:

(i) long-lived birds and bats with high metabolic rates reduce the damaging impact of ROS by a modified mitochondrial membrane and an increased defence against protein damage [60,61]; and

(ii) the very long-lived and disease-resistant naked mole rat displays increased intrinsic oxidation resistance of its proteome via an approximately 10-fold reduction in translational error rate [62] (fig. 2 in [1]) and an increased structure-based protein stability [63]—both correlating with protein oxidation resistance [1].

Thus, a generic 'elixir' for healthy longevity should support high ATP production along with low ROS-mediated protein damage. Reviewing figures 1, 2, 4 and 5 recapitulates two basic concepts for ARD and ageing proposed in this and the companion paper [1]: the first is at molecular (fig. 3 in [1]) and the second at the cellular/tissue (figures 2 and 4) level. Together, they could interpret the basic phenomenology of ageing and ARD (figure 5).

# 4. Daydreaming of a new medicine

Monumental human and financial resources, unequalled in the history of science and technology, have been invested worldwide in over half a century of biomedical research with dismal success in its mission to mitigate cancer and other ARD. It becomes a matter of ethics to cast doubts whether the mainstreaming of biomedical research (through funding and publication criteria) to 'mechanistic studies' of the consequences of ageing and ARD is adequate for the task. It is legitimate to dream of a better medicine based on prevention rather than therapy and on more relevant research such as to identify and act upon the root cause(s) of ARD. Currently, the research on active prevention of diseases, equivalent to vaccination, is practically neglected in favour of therapies of advanced diseases that may never succeed with the 'war on diseases' approach because ARD reside within our own cells. Perhaps a 'biomimetic' approach would be more productive since evolution has already found ways to reduce initiation and promotion of diseases in long-lived animals. In such a case, we should support free academic research providing basic knowledge necessary for prevention and healing of ARD. Instead, we have to justify fundamental research because of pretended short-term lack of utility while monumental funding continues for mission-oriented research that showed its long-term lack of utility.

Validation of the concepts proposed in the two companion papers, just on common ARD, risks taking a long time. But what if they are basically correct? Obvious potential applications were mentioned throughout this and the companion paper [1], such as (i) prevention of diseases by pharmacological optimization of cellular parabiosis to extend disease latency, or to re-establish parabiosis with healthy cells to heal a developing disease by reversion; (ii) interruption of cellular parabiosis within tumours may ameliorate tumour therapies by weakening tumour cell fitness (in both cases via a stringent control of the relevant part of inflammation response, e.g. via an adequate inhibition of matrix proteases); (iii) prevention of oxidative damage to proteins and stimulated sanitation of oxidatively damaged intra- and extracellular proteomes [1]; and (iv) targeted defence against oxidation of susceptible proteoforms that predispose to particular disease [1]. Pharmacological prevention and reversion of early stage ARD by enhancing natural mechanisms is a rational alternative to current treatments of disease symptoms. This is a dreamed-up version of a personalized preventative and curative medicine.

Data accessibility. This article has no additional data.

Competing interests. We declare we have no competing interests.

Funding. This research was funded by Fondation Jean-Noël Thorel and the NAOS Group and by the Swiss Fondation Nelia et Amadeo Barletta (2011–2014). We acknowledge the funding in 2018, by DC Europa Ltd (Medicxi) Cambridge, UK.

Acknowledgements. The author is grateful to his friend Jean-Noël Thorel, as well as to Mr Branko Roglic (Orbico, dd), for sponsoring the research on biological robustness, ageing and diseases in MedILS. The author enjoys a long collaboration with Anita Krisko and enlightening discussions with three extraordinary MDs: Philippe Even, Zoran Dermanovic and David Grainger.

This paper was inspired by, and is dedicated to, the memories of two teachers and friends: Max Perutz believed in the power of simple concepts in exploring complexity, and Bernard I. Weinstein taught me biology of cancer in patients versus in Petri dishes. The trigger to write was the correspondence with George Klein in 2015. Harry Rubin's review [21] was a revelation. Remembering conversations with Michael Stocker, John Cairns, Richard Peto, Matt Meselson, Errol Friedberg and Michael Brown impacted this paper. Any

mistakes are the author's. The drawings are by Mikula Radman and Maria Topalovic.

A spectacular demonstration of cellular parabiosis shows in plant cells a massive presence of proteins encoded by genes residing in other cells of the same tissue. Such functional complementation at supracellular level allows for the reproduction of a multipartite virus with genome fractions present in different cells [64].

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
